# The Parameters Identification of High-Temperature Constitutive Model Based on Inverse Optimization Method and 3D Processing Map of Cr8 Alloy Steel

**DOI:** 10.3390/ma14092216

**Published:** 2021-04-26

**Authors:** Xuewen Chen, Tingting Lian, Bo Zhang, Yuqing Du, Kexue Du, Bingqi Liu, Zhipeng Li, Xuanhe Tian, Dong-Won Jung

**Affiliations:** 1School of Materials Science and Engineering, Henan University of Science and Technology, 263 Kaiyuan Avenue, Luoyang 471023, China; 141402060513@stu.haust.edu.cn (T.L.); 190319020161@stu.haust.edu.cn (B.Z.); 180318020132@stu.haust.edu.cn (Y.D.); 180318020122@stu.haust.edu.cn (K.D.); lbq8565@stu.haust.edu.cn (B.L.); 181402060810@stu.haust.edu.cn (Z.L.); 181402060223@stu.haust.edu.cn (X.T.); 2Faculty of Mechanical, Jeju National University, Jeju Island 63243, Korea

**Keywords:** Cr8 alloy steel, Hansel–Spittel constitutive model, inverse optimization method, 3D thermal processing map, thermal deformation behavior, high temperature compression test

## Abstract

As a novel kind of cold roller steel, Cr8 alloy steel has the characteristics of high hardness, high wear resistance and good toughness, which can effectively prolong the service life of the roller that is an important part of the steel rolling mill. How to accurately define the constitutive model parameters of metal materials is the major problem, because it seriously affects the accuracy of numerical simulation results of the roller hot forming process. In the study of Cr8 alloy steel’s thermal deformation behavior of the present paper, the high temperature compression test was done on a Gleebel-1500D thermal/force simulation testing machine. A novel method of parameter identification was proposed based on inverse optimization. The Hansel–Spittel constitutive model was established by using the inverse optimization method. To carry out the verification on the accuracy of the established constitutive model, the predicted flow-stress of constitutive model was made a contrast to the experimental flow-stress, and the standard statistical parameters were also applied to further evaluation. The results showed a relatively high prediction accuracy of the Hansel–Spittel constitutive model based on the inverse optimization algorithm. Meanwhile, to obtain optimal parameters of Cr8 alloy steel in the thermal processing, 3D thermal processing maps concerning strain-rate, strain and temperature were built based on the dynamic material model. According to the 3D processing map, the most adequate thermal processing parameters of Cr8 alloy steel were obtianed as follows: strain 0.2–0.4, strain-rate 0.05–0.005 s^−1^, temperature 1100–1150 °C.

## 1. Introduction

Cr8 alloy steel is a new kind of cold roller steel, which has the characteristics of high hardness, high wear resistance, high heat cracking resistance, good hardenability and toughness. As a result, it can effectively improve the service life of the roller, which is an important part of the steel rolling mill. Cr8 alloy steel is forged to obtain good mechanical properties. In order to produce qualified rolling mill rolls, it is required that Cr8 alloy steel has good plasticity. Therefore, we need to change the microstructure of Cr8 alloy steel through thermal deformation to improve its processing performance. However, at present, the research on the properties of Cr8 alloy steel has been not very profound, and it is difficult to determine the best parameters of the production process. It has been still in the trial-and-error production stage so far, which is difficult to meet the output and performance requirements of large forgings. The constitutive model of material is often used to quantitatively study the thermal deformation behavior. The major issue seriously affecting the accuracy of the numerical simulation results of roll in the hot forming process is the inaccurate identification of material constitutive model parameters. Therefore, it is necessary to establish an accurate constitutive model to make an intensive study of the thermal deformation behavior of Cr8 alloy steel and optimize the process parameters to improve the production efficiency.

Until now, constitutive models have mainly been divided into the empirical model, phenomenological model and physical model [1]. Among them, the phenomenological constitutive model has fewer material constants and is easy to be calibrated [2]. After intuitively grasping the direct influence of strain, strain-rate and thermal deformation temperature on flow-stress, a phenomenological constitutive model was able to be built, using a regression method according to experimental data [3]. Nowadays, scholars have established many phenomenological models, such as Arrhenius model, Zerilli–Armstrong model, Johnson-Cook (JC) model, Fields-Backofen (FB) model, Hansel–Spittel constitutive model and so on [3]. Liu Yue [4] used the Arrhenius model to describe the thermal deformation behavior of Q690 V-N and V-N-Cr steel, and the correlation coefficients were 0.999 and 0.987, respectively. Olga Yakovtseva [5] used the mathematical Arrhenius constitutive model and artificial neural network model to predict the flow behavior of Al-Zn-Mg alloy, and the results showed that both of them had good predictability, and the prediction accuracy of the artificial neural network model was slightly higher. Shuai He [6] took the effect of strain on the Arrhenius model′s constants and thermal deformation activation energy (Q) as an independent function, and used the sixth-degree polynomial to perform the fitting. The correlation coefficient (R) and average absolute relative error (AARE) of the measured and predicted data were estimated to be 0.99026% and 3.94%, respectively, which verified the precision of the modified Arrhenius constitutive model. The model could precisely describe the flow-stress constitutive relation of materials. However, when studying the effect of strain variables on the flow-stress, further numerical fitting requires a large amount of work, and there was also a large prediction error. A constitutive model was proposed by Hensel and Spittel (1978) to represent the flow-stress of rigid plastic materials [7]. What is more, the Hansel–Spittel equation had a better prediction accuracy than the Arrhenius equation in the strain compensation on the basis of the thermal tensile deformation of Ti-6Al-4V alloy [8]. The Hansel–Spittel constitutive model, which considers the effects of strain, strain-rate and temperature on the metal flow-stress, can describe the thermal deformation behavior of materials more precisely. Qiang Liang [9] proposed a high temperature Hansel–Spittel constitutive model, which laid a foundation for theoretical and engineering analysis of HNi55-7-4-2 alloy. Therefore, the Hansel–Spittel constitutive model was used to make a prediction on the high temperature thermal deformation behavior of Cr8 alloy steel.

The process of deducing the constitutive model parameters by the traditional mathematical method is complicated, and the parameter values of the constitutive model obtained by the average method have a bad effect on predicting the flow-stress accurately. For example, Li Xue-song [10] modeled the shape of the compression curve by calculating the parameters of six simultaneous equations containing 10 different parameters. The whole process of Hansel–Spittel model was very complicated [10]. Haoran Wang established strain compensating the Hansel–Spittel model to get a better description on the behavior of thermal deformation of 20Cr2Ni4A alloy steel during warm and high temperature deformation [4]. However, the Hansel–Spittel constitutive model has nine parameters. It is difficult to derive the model parameters by using traditional mathematical methods, and the parameters of various deformation conditions obtained by the evaluation means will reduce the accuracy in the constitutive model prediction. Therefore, a method that can accurately identify the parameters of the Hansel–Spittel constitutive model is needed to analyze the thermal deformation behavior of materials and finite element simulation. So far, few scholars have combined the inverse optimization algorithm with the Hansel–Spittel constitutive model and used the optimization inverse method to determine the parameters of the constitutive model. The inverse optimization algorithm minimizes the value of the error function. The error function is the difference between the overall data obtained in the hot compression experiment and the corresponding parameters set in the optimization experiment [11]. M. Abbasi [12] proposed a new method for the inverse calculation of GTN plastic damage model parameters by using the response surface optimization algorithm. The results showed the predicted data were very consistent with the experimental data. Giovanni B. Broggiato [13] used the inverse method to accurately determine the parameters of the damage model. Sylvain Charles [14] proposed a new method for determining constitutive antithesis parameters based on total kinematics and thermal field measurements. It could be seen that there was a smaller parameter error and higher prediction accuracy in the constitutive model obtained by the inverse optimization algorithm.

The processing map plays a critical part in describing the processability of the material. Prasad et al. [15] first proposed the processing map based on the Dynamic Material Model (DMM) theory. The processing map not only reflects the deformation characteristics of different evolutionary mechanisms, but also shows the rheological instability areas during the hot forging process, thus obtaining the optimal hot forging temperature and strain rate. Since the traditional 2D processing map does not contain strain, the influence of strain on the hot forging of materials cannot be clearly expressed. From the hot forging of materials perspective, we built a 3D processing map including strain because strain can influence it greatly as shown in Shalbafi [16], who established a 2D processing diagram of Mg-10Li-1Zn alloy at a high temperature and different strains. Madlen Ullmann [17] established a 2D processing map of Mg-6.8Y-2.5Zn-0.4Zr Magnesium Alloy to identify the unstable areas in the material and found that the adiabatic shear band or the instability of plastic flow had a tendency to occur during deformation. Enxiang [18] established 2D processing maps of superaustenitic stainless steel S32654 at different temperatures and strains. Results showed that it had difficulty in obtaining the instability parameter overall variation law and power dissipation efficiency directly. The 3D processing maps of AZ31B were built by Liu et al. [19]. The 3D processing maps of Ti-47Al-2Nb-2Cr alloys were established by Sun et al. [20], using a temperature coordinate, strain-rate coordinate and strain coordinate. The results expressed that the 3D processing map could accurately describe the thermal deformation behavior and microstructure evolution and provide the optimized hot forging parameters. The conventional processing map is independent of strain. Liu [21] constructed 3D processing maps concerning the effect of strain on machinability, and studied the metal machinability of magnesium alloy ZA31B by making the 3D processing maps. Wang [22] confirmed the stable and unstable processing conditions during the whole deformation process and determined the optimal processing parameters of AA7050 aluminum alloy by combining the traditional 2D and 3D processing maps. Few researchers studied the deformation behavior of materials by using 3D processing map in the past. At present, there have been few studies on thermal deformation characteristics and thermal machining performance design by using a 3D processing map. Until now, there have not been reports on the processing map of Cr8 alloy steel, especially the 3D processing map. Therefore, its thermal deformation behavior is worthy of in-depth study.

Therefore, the thermal deformation behavior of Cr8 alloy steel was researched through high temperature hot compression tests in the present paper. A novel inverse optimization identification method of Hansel–Spittel constitutive model parameters was proposed. There was a further evaluation on the precision of the constitutive model by contrasting the predicted flow-stress with the test flow-stress by using the standard statistical parameters. For the purpose of optimizing the process parameters, a 3D processing map considering strain, strain-rate and temperature was set up based on the dynamic material model.

## 2. Materials and Experimental Procedure

Cr8 alloy steel was applied as the material in the paper, and its chemical composition (mass fraction, wt.%) has been expressed in Table 1. The experimental material is rod material annealed at 650 °C. The test sample was processed by wire cutting to a cylindrical shape of Ø 8 × 12 mm^2^. The thermal compression test was carried out on the Gleeble-1500D (Dynamic Systems, New York, USA) thermal simulation experimental machine. The test deformation temperatures were 900, 975, 1050, 1125 and 1200 °C, and the deformation rates were 0.005, 0.01, 0.1, 1 and 5 s^−1^. Graphite flake lubricant was applied to two ends of the sample to ensure minimum friction during hot compression. Before the hot compression, each sample was heated at 10 °C/s to the set test temperature and stayed for 5 min to ensure that the temperature of all parts had no differences so that there is no elimination of the thermal temperature gradient inside the material. Then, the compression deformation happened according to the set compression rate until the true-strain was 0.7. In order to keep the deformation state under the corresponding deformation condition that is convenient for the observation of metallographic structure, the samples were quenched to room temperature with water for less than 2s when the deformation amount reached 50%. The test process scheme has been shown in Figure 1. The deformed Cr8 alloy steel sample was cut into two halves by the wire cutting method and its microstructure was observed and analyzed. The sample for the optical metallographic test was polished by an abrasive paste and etched with an etching solution prepared with picric acid.

## 3. Flow Curve of Cr8 Alloy Steel High Temperature Compression Test

The typical flow-stress curve of Cr8 alloy steel in the condition of disparate deformation has been shown in Figure 2. From Figure 2, we can see the flow-stress in the range of study increases with the decrease of the deformation temperature while it increases with the corresponding strain-rate increasing during the thermal compression process. The evolution of flow-stress with strain has three different phases—work hardening, softening and stability [23]. In the stage of micro-strain, the flow-stress linearly rises with the strain increasing, and that is primarily because of the continuous proliferation of dislocation density with the increase of deformation degree, resulting in work hardening and the accumulation of deformation energy. As the work hardening rate increases, the deformation resistance will continue to increase until it reaches the peak value, and then the rate of the curve goes down. The primary reason can be attributed to dynamic softening mechanisms (e.g., DRV and DRX), that is, work hardening will be counteracted or partially counteracted. When the effect of softening is stronger than the strain hardening, then the curve may show a descending trend due to the sufficient DRX behavior. As shown in Figure 2b, in the flow curve at 900 °C, when the strain is 0.25, the maximum stress value is 260.49 MPa. After that, with the rise of true-strain, the true-stress slowly decreases, and the final true-stress value remains at about 270.84 MPa, which keeps stable with the change of the true-strain. This curve can be classified as a dynamic recrystallization. As the deformation increases, the stress has a trend to be a stable value and the deformation enters the third stage (stable stage). This is caused by work hardening and dynamic softening phase equilibrium when the deformation reaches a certain stage. As shown in Figure 2b, the flow curve at 1200 °C rises rapidly at first, then slows down with the rise of strain, and finally reached the maximum stress value of 85.82 MPa, which no longer changes with the change of true-strain. Such a curve is called a typical dynamic re-recovery curve.

Figure 3 describes the microstructure of Cr8 alloy steel after thermal compression deformation. We can see from the Figure 3 that at lower temperature and lower strain-rate, smaller grains are easily obtained. The corresponding deformation conditions in Figure 3a are a temperature of 975 °C and strain-rate of 0.005 s^−1^, and the average grain size obtained is 6.05 μm. Because of the low deformation temperature, low strain-rate and large grain refinement, the dynamic recrystallization grains almost cover the whole original coarse grain, and the dynamic recrystallization structure is relatively uniform. The corresponding deformation conditions in Figure 3c are a temperature of 1125 °C and strain-rate of 0.01 s^−1^, and the average grain size obtained is 14.15 μm. Compared with those in Figure 3a,c, the grain size is relatively large, indicating that dynamic recrystallization can be developed fully when the temperature increases. The corresponding deformation conditions in Figure 3b are a temperature of 975 °C and strain-rate of 0.1 s^−1^. The average grain size under these conditions is 2.72 μm. Many shear bands appear in the figure, resulting in uneven mechanical properties, and these shear bands extend from one side of the upper part to the other side of the lower part of the sample, and the strain concentration is more serious at the site where they occur. Aiming at obtaining better mechanical properties, such processing conditions should be avoided in actual production.

The Cr8 alloy steel’s peak-stress under the disparate deformation conditions has been demonstrated in Figure 4. Figure 4 expresses that during the thermal deformation process, Cr8 alloy steel’s peak-stress is very susceptible to be affected by the deformation temperature as well as the strain-rate. In the case of a fixed testing strain-rate, with the deformation temperature increasing, the peak-stress decreases and its decreasing rate decreases slightly. The peak-stress will increase with the increase of strain-rate in the case of a fixed testing temperature, having a significant decrease in its increasing rate. This indicates that when the value of strain-rates is high, there is less time on the dynamic softening mechanism to integrate energy and reduce dislocation density, leading to a higher dislocation accumulation degree. As a result, the peak-stress is obviously lower at low strain-rate compared with the value at a high strain-rate. At a high temperature (1200 °C), the peak-stress of Cr8 alloy steel increases from 34.87 to 117.11 MPa with an increase of 235.8%, with the strain-rate increasing. At a low strain-rate (0.005 s^−1^), when the temperature of deformation increases, the peak-stress of Cr8 alloy steel decreases from 155.46 to 34.87 MPa with a decrease of 77.6%. The lowest strength of the material in thermal deformation can be obtained by reducing the strain-rate as well as by increasing the temperature of deformation. This is because work hardening in the condition of low temperature or high strain-rate is the material’s main deformation. Either by increasing the temperature or decreasing the strain-rate, the material’s flow-stress can be reduced, so the deformation time and energy are sufficient for dynamic recovery and recrystallization.

## 4. The Parameters Identification of the Hansel–Spittel Constitutive Model Using Inverse Optimization

To study the plastic flow characteristics of Cr8 alloy steel during hot deformation, it is necessary to carry out constitutive analysis. The constitutive model can be used for the description on the quantitative relationships of the hot deformation’s flow-stress, strain, strain-rate and deformation temperature. Constitutive relation is the premise of numerical simulation of macroscopic and microstructural changes in the metal plastic machining process by using the rigid plastic finite element method. It is also the main basis of reasonable formulation of various process parameters and accurate selection of equipment in the deformation process. The Hansel–Spittel constitutive model quantifies the effect of strain, temperature and strain-rate on the flow-stress, which has applicability to wide applications and can present the thermal deformation behavior of Cr8 alloy steel more accurately. Moreover, the commercial finite element software FORGE integrates the Hansel–Spittel Constitutive Model, and identifying the constitutive model parameters can provide the premise for the subsequent finite element simulation. The Hansel–Spittel constitutive model has been shown in Equation (1):(1)$$\sigma =A\cdot e^{m_{1}T}\cdot \varepsilon^{m_{2}}\cdot {\dot{\varepsilon }}^{m_{3}}\cdot e^{\frac{m_{4}}{\varepsilon }}\cdot {\left(1+\varepsilon \right)}^{m_{5}T}\cdot e^{m_{7}\varepsilon }\cdot {\dot{\varepsilon }}^{m_{8}T}\cdot T^{m_{9}}$$
where, *A* is the material constant; *m*_1_ and *m*_9_ define the material′s sensitivity to temperature; *m*_2_, *m*_4_, and *m*_7_ define the material′s sensitivity to strain; *m*_3_ depends on the material′s sensitivity to the strain rate; *m*_5_ terms coupling temperature and strain; *m*_8_ terms coupling temperature and strain rate.

Since the constitutive model has many parameters and the traditional mathematical method is also cumbersome to solve and the accuracy is low, it can be considered to introduce the optimization method to obtain the parameters of the Hansel–Spittel constitutive model of Cr8 alloy steel.

The genetic algorithm (GA), as a method to search the optimal solution by simulating the natural evolution process, has good global properties, but it tends to converge prematurely, and its solving efficiency is lower than other optimization methods. The mountain climbing algorithm selects an optimal solution from the adjacent solution space of the current solution as the current solution every time until a local optimal solution is reached. Its main disadvantage is that it will fall into the local optimal solution. One of the most appropriate ways to solve nonlinear optimization problems is Sequential Quadratic Programming (SQP) [24]. It has the characteristics of excellent convergence, high computational efficiency, and strong boundary search abilities [24]. It can effectively solve nonlinear least squares and constraint problems based on numerical experiments [24]. Therefore, an optimization algorithm has been used to calculate the parameters in this paper. Firstly, each parameter to be solved is defined, and then initial values are assigned to the unknown parameters of the Hansel–Spittel constitutive model, and the SQP optimization algorithm is selected as the identification strategy to minimize the goal function. Finally, the errors between the divinable value and the experimental value of flow-stress can be evaluated by the objective function, and the parameter corresponding to the minimum error is the optimal solution. The optimization process has been shown in Figure 5 and the selected objective function has been shown in Formula (2).
(2)$$\varphi =\frac{\sum_{i}\left[{\left(y_{i}{}^{\exp}-y_{i}{}^{num}\right)}^{2}\left(x_{i}-x_{i-1}\right)\right]}{\sum_{i}\left[{\left(y_{i}{}^{\exp}\right)}^{2}\left(x_{i}-x_{i-1}\right)\right]}$$

In the formula, $y_{i}{}^{\exp}$ represents the stress belonging to the *i*th data point of the test, $y_{i}{}^{num}$ represents the stress belonging to the predicted *i*th data point, $x_{i}$ represents the *i*th data point, and $x_{i-1}$ represents the *i*−1th data point. The closer the objective function φ is to 0, the closer the prediction result is to the test result.

Finally, according to the Hansel–Spittel constitutive model and the set objective function, the optimal parameters were obtained based on high temperature compression experimental data, as presented in Table 2.

By applying the material constant calculated above to Equation (1), the predicted stress value under the whole experimental condition in this study could be obtained. As it shown in Figure 6, to validate the accurate prediction of the established constitutive model, the predicted flow-stress was contrasted with the experimental value, the data have shown that there had an excellent consistency between the calculated and the experimental results.

A small number of true stress-strain curves have large differences between the experimental and predicted values, this may be due to the fact that the dynamic softening effect is completely different from others, and the Hansel–Spittel constitutive model tends to describe most real stress-strain conditions.

Standard statistical parameters such as Root Mean Square Error (*RMSE*), Correlation Coefficient (*R*) and Average Absolute Relative Error (*AARE*) can also be used for further evaluation of the constitutive model [25]. They can be stated as follows:(3)$$R=\frac{\sum_{i=1}^{N}\left(\sigma_{e}^{i}-{\bar{\sigma }}_{e}\right)\left(\sigma_{e}^{i}-{\bar{\sigma }}_{p}\right)}{\sqrt{{\sum_{i=1}^{N}\left(\sigma_{e}^{i}-{\bar{\sigma }}_{e}\right)}^{2}}\sqrt{{\sum_{i=1}^{N}\left(\sigma_{e}^{i}-{\bar{\sigma }}_{p}\right)}^{2}}}$$
(4)$$RMSE=\sqrt{\frac{1}{N}{\sum_{i=1}^{N}\left(\sigma_{e}^{i}-\sigma_{p}^{i}\right)}^{2}}$$
(5)$$AARE\left(\%\right)=\frac{1}{N}\sum_{i=1}^{N}\left|\frac{\sigma_{e}^{i}-\sigma_{p}^{i}}{\sigma_{e}^{i}}\right|\times 100$$

In the formula, $\sigma_{e}$ is the testing stress data, $\sigma_{p}$ is the predicted stress data. ${\bar{\sigma }}_{e}$ is the average value of $\sigma_{e}$ and ${\bar{\sigma }}_{p}$ is the average value of $\sigma_{p}$. *N* (2500) is the total amount of data.

The R has a reflection on the linear relationship between two sets of data. However, higher R values sometimes do not necessarily mean better predictability and reliability, constitutive models may favor lower or higher values. In addition, RMSE is the standard error, and AARE can be regarded as an unbiased statistic to evaluate the performance of a built model [26]. Therefore, when RMSE and AARE are low enough, the prediction performance of the constitutive model will be very good. The calculated results were 0.999 (R), 9.64 MPa (RMSE) and 8.09% (AARE), respectively. Therefore, the established constitutive model has excellent ability to forecast the flow-stress of Cr8 alloy steel at a high temperature.

## 5. 3D Hot Processing Map of Cr8 Alloy Steel

To confirm the optimal thermal processing parameters of Cr8 alloy steel, understanding the plastic processing performance of the material under different conditions is a necessity. The processing diagram can express the advantages and disadvantages of the processing performance of the material itself. Establishing a processing map on the basis of a dynamic material model (DMM) is a significant part for the optimization on parameters of thermal processing and the control on the microstructure [27]. Based on the theory of DMM [28], the power loss (P) during thermal deformation of the workpiece consists of two non-separate parts—the power dissipated by plastic processing (*G*) and microstructure evolution (*J*), as follows [29]:(6)$$P=\sigma \dot{\varepsilon }=G+J=\int_{0}^{\dot{\varepsilon }}\sigma d\dot{\varepsilon }+\int_{0}^{\sigma }\dot{\varepsilon }d\sigma$$
where σ is the stress, $\dot{\varepsilon }$ is the strain-rate, *G* is the power dissipated parameter, and *J* is the power dissipated parameter.

At the conditions of the given strain and temperature, flow-stress has been shown as follows:(7)$$\sigma =K{\dot{\varepsilon }}^{m}$$

In this case, *K* is a constant, m is the coefficient of strain-rate sensitivity calculated by the Formula (8) [30]:(8)$$m=\frac{dJ}{dD}=\frac{\dot{\varepsilon }d\sigma }{\sigma d\dot{\varepsilon }}=\frac{d\ln\sigma }{l\ln\dot{\varepsilon }}$$

Then we can get the power consumption efficiency (*η*) from m:(9)$$\eta =\frac{2m}{m+1}$$

In addition, an instability criterion established by Prasad et al. [15] for judging the unstable area based on DMM, showed by the flow instability parameter:(10)$$\xi (\dot{\varepsilon })=\frac{\partial \log\frac{m}{m+1}}{\partial \log\dot{\varepsilon }}+m<0$$

Figure 7 has shown that Cr8 has three peak power consumption zones—when the value of strain is 0.3, the power dissipation factor reaches the peak value of 40% in the range of temperatures 1100–1150 °C and strain-rates 0.05–0.005 s^−1^; When it is 0.5, the factor achieves the peak value of 33% in the range of 950–975 °C and 0.05–0.018 s^−1^; When it is 0.5, the factor reaches a peak value of 38% in the range of 1175–1200 °C and 0.5–0.05 s^−1^. The instability mostly occurs in the area where the strain-rate is high while temperature is low [31], because the dynamic recrystallization is not sufficient at low temperatures and high strain-rate as well as the hot workability is poor [32].

The 2D hot processing map represents the machinability of the workpiece under the condition of determined strain. However, the strain distribution of the workpiece in actual production is very uneven, so it is difficult to analyze the machinability of the workpiece effectively with the 2D hot processing map.

To see the power dissipation coefficient’s evolution and the rheological instability area more clearly and to determine the appropriate processing area, a 3D processing map concerning strain has been established. Based on the instability coefficient and power dissipation factor coefficient obtained above, a 3D coordinate axis was taken with strain, strain-rate and temperature [33]. Figure 8 shows the power dissipation map and instability map, both in 3D.

Through power consumption efficiency (*η*) changing with the temperature and test strain-rate, the map of power dissipation was constituted, as shown in Figure 8a. The color of the cloud image represents the percentage of power efficiency. As seen from Figure 8a, the strain-rate has a huge impact on the power dissipation map. As the strain increases, the power dissipation efficiency changes in a large range. This is because the stored energy will increase as the dislocation accumulates when strain increases [29], however, it may be partially consumed by the movement of the dislocation or changes in the microstructure (e.g., dynamic recovery and recrystallization). The area of high dissipation efficiency gradually broadens with increasing temperature. With the temperature increasing, the power dissipation increases. At first, the power dissipation in the medium and high temperature regions increases and then decreases. What is more, the power dissipation in the low temperature region increases with the decrease of strain-rate. There are two power dissipation peak areas in the Cr8 alloy steel processing map. The first peak region is a temperature 1100–1150 °C, strain rate is 0.05–0.005 s^−1^; the second peak region’s temperature is 950–1000 °C, strain rate is 0.005–0.05 s^−1^. The microstructure shown in Figure 3a (microstructure at temperature of 975 °C and strain rate of 0.005 s^−1^) has fine grains and good machinability. It was obtained by deformation in the second power dissipation peak region.

The instability diagram drawn by the instability criterion proposed by Prasad [34] has been shown in Figure 8b. The red area represents the flow instability area. Figure 8b shows the instability area. The instability zone of Cr8 alloy steel concentrates in the range of strain-rate 0.1–5 s^−1^ and temperature 900–1125 °C, and with the strain increasing, the instability zone reduces first and then expands. As can be seen, with the increase of strain rate, strain and temperature, the instability zone of Cr8 alloy steel expands. That is, at the higher strain-rate, the grains grow rapidly and form new recrystallized ones at boundaries of the grains, resulting in uneven grain structure [35]. Therefore, these instability zones should be prevented in the hot processing. Many shear bands (microstructure at a strain rate of 1 s^−1^ at a temperature of 975 °C) shown in Figure 3b have poor microstructure uniformity, which were obtained by deformation in the instability zone.

The region without instability is the safety region. In this region, when the power dissipation coefficient is greater, the thermal working performance of the material will be better. Therefore, the most suitable processing parameters of Cr8 alloy steel are as follows—strain-rate 0.05–0.005 s^−1^, temperature 1100–1150 °C.

## 6. Conclusions

In this study, Cr8 alloy steel’s hot deformation behavior was analyzed in thermal compression tests under widely ranging deformation temperatures (900–1200 °C, strain-rate 0.005–5 s^−1^). The major conclusions drawn are as follows.

The Cr8 alloy steel’s flow-stress in a wide range of 900–1200 °C shows two characteristics. One is that with the strain increasing, the flow-stress has a slow rise, and the true-stress basically shows a relatively steady condition after it reaches a peak, indicating dynamic recovery characteristics. The other is that the flow-stress will have a rapid rise along with the strain increasing, reaching a peak, then goes down to a stable state, showing typical dynamic recrystallization characteristics.

A new Hansel–Spittel constitutive model parameters inverse optimization method was proposed in this paper. By comparing the measured values of the experiment and the predicted values of the constitutive model, the objective function was set to a minimum to solve the model parameters. The Hansel–Spittel constitutive model established based on the reverse optimization algorithm has very accurate prediction performance.

The 3D hot processing maps have been established according to the dynamic material model (DMM), which showed that Cr8 alloy steel was most suitable for processing in the range—strain 0.2–0.4, strain-rate 0.05–0.005 s^−1^, temperature 1100–1150 °C.

## Figures and Tables

**Figure 1 materials-14-02216-f001:**
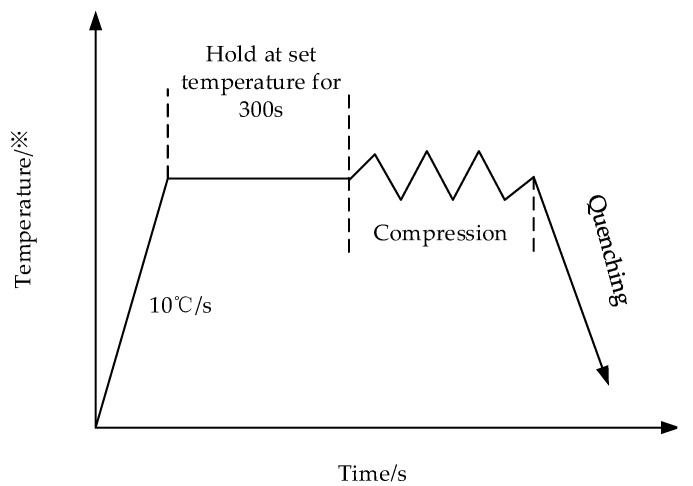
Test process scheme of Cr8 alloy steel.

**Figure 2 materials-14-02216-f002:**
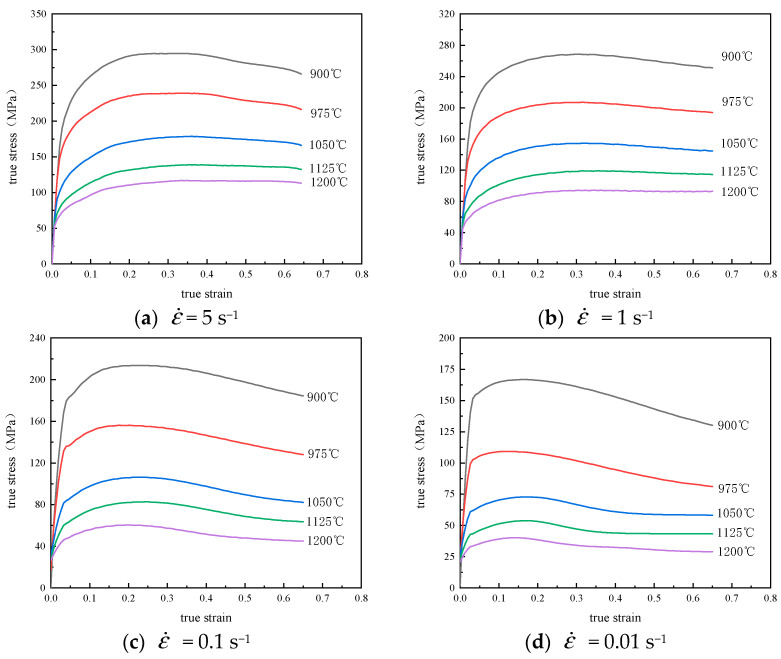
Cr8 alloy steel’s true-stress-strain curves.

**Figure 3 materials-14-02216-f003:**
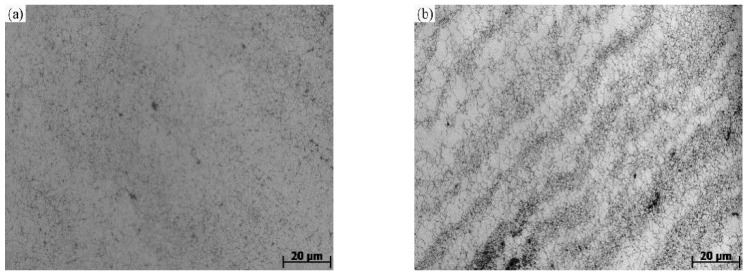
Microstructure of Cr8 alloy steel at different temperatures and strain-rates. (**a**) 975 °C, 005 s^−1^; (**b**) 975 °C, 1 s^−1^; (**c**) 1125 °C, 0.01 s^−1.^

**Figure 4 materials-14-02216-f004:**
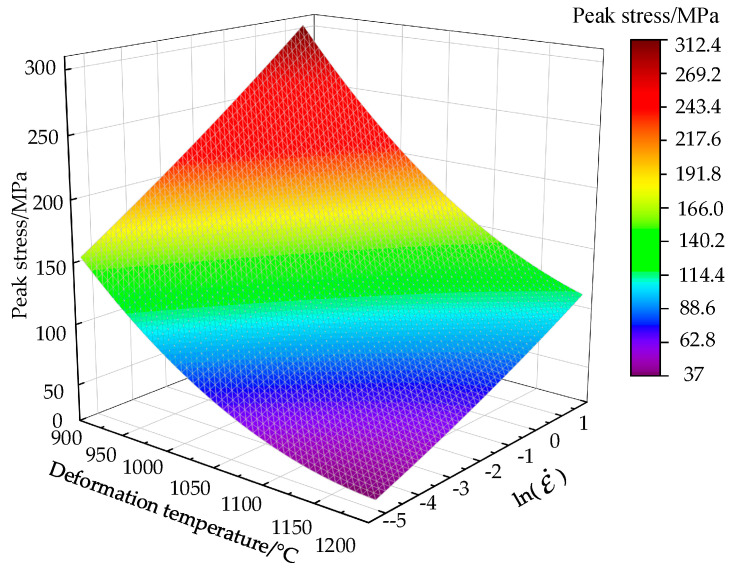
Variation of peak-stress of Cr8 alloy steel at various deformation conditions.

**Figure 5 materials-14-02216-f005:**
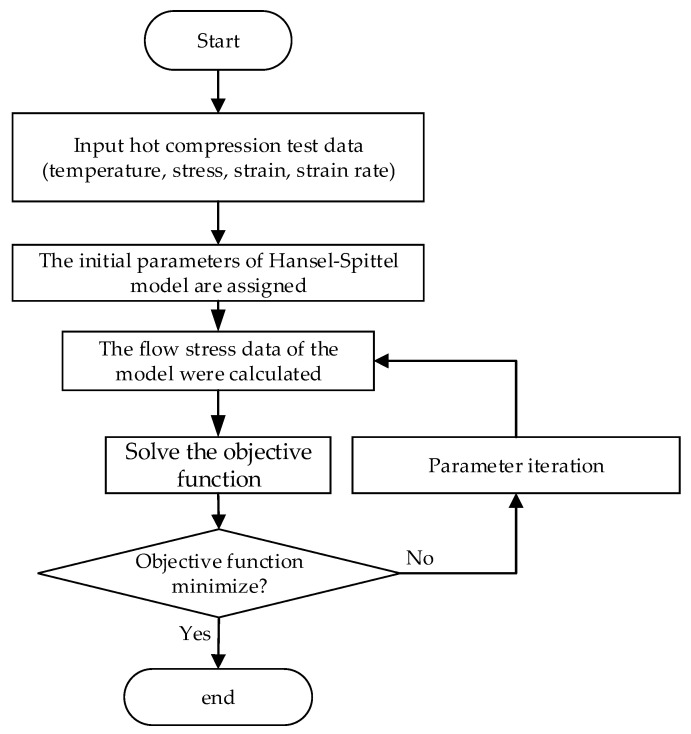
Parameter optimization process.

**Figure 6 materials-14-02216-f006:**
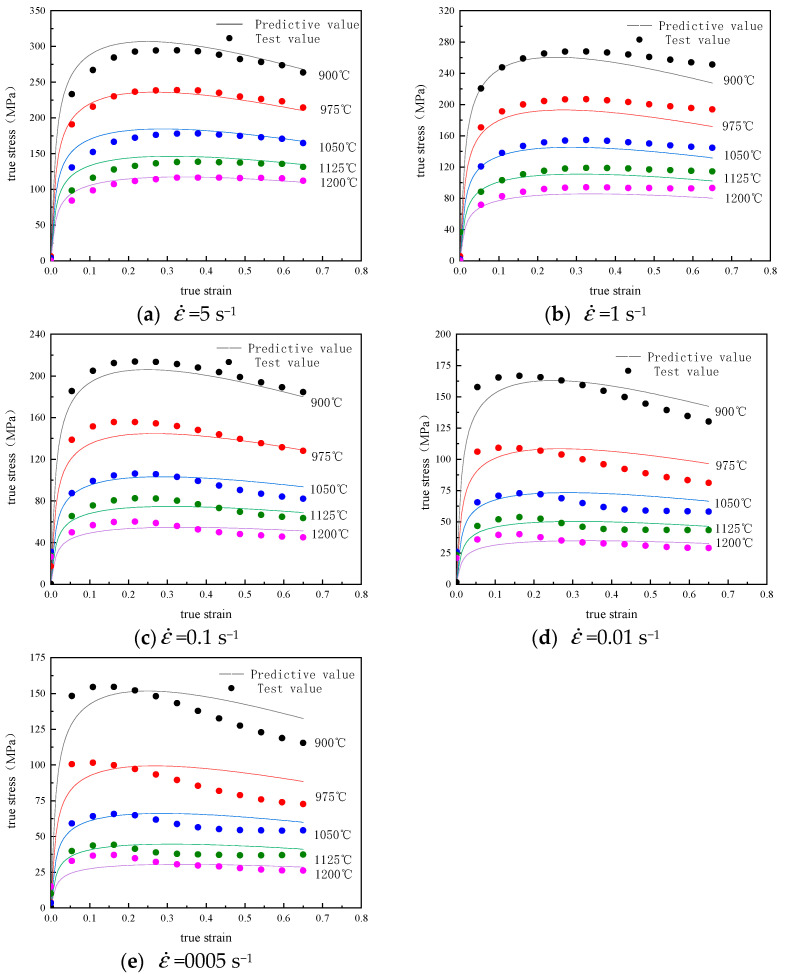
Comparison of experiments and predicted flow-stresses at various strain-rates.

**Figure 7 materials-14-02216-f007:**
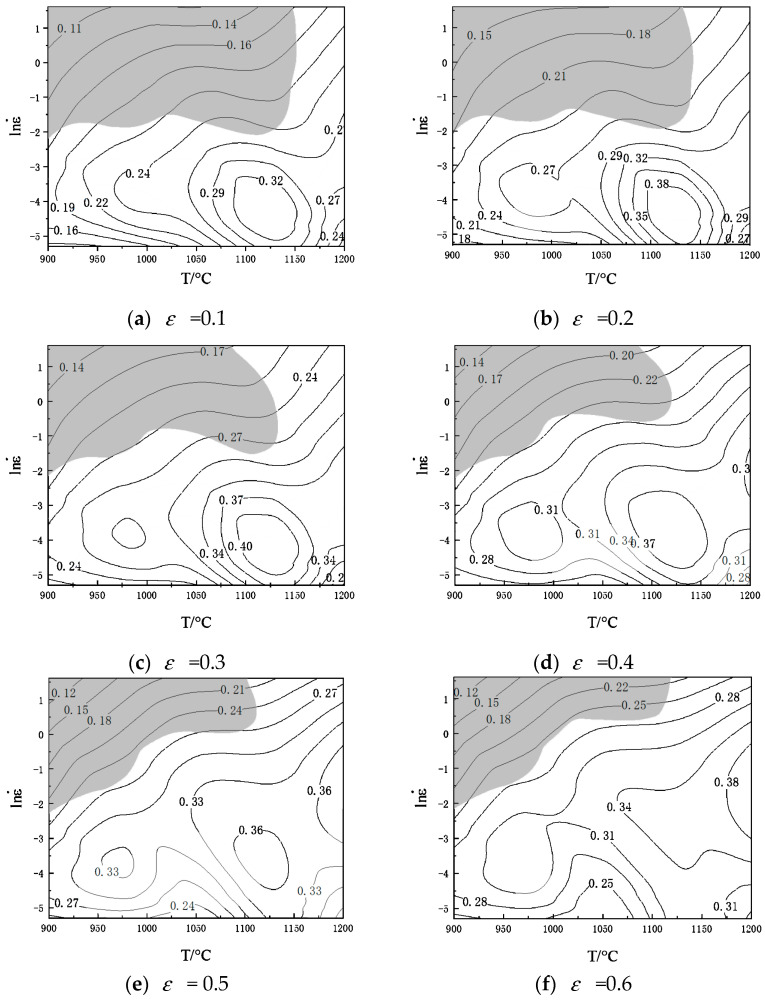
The 2D hot processing maps at various strains.

**Figure 8 materials-14-02216-f008:**
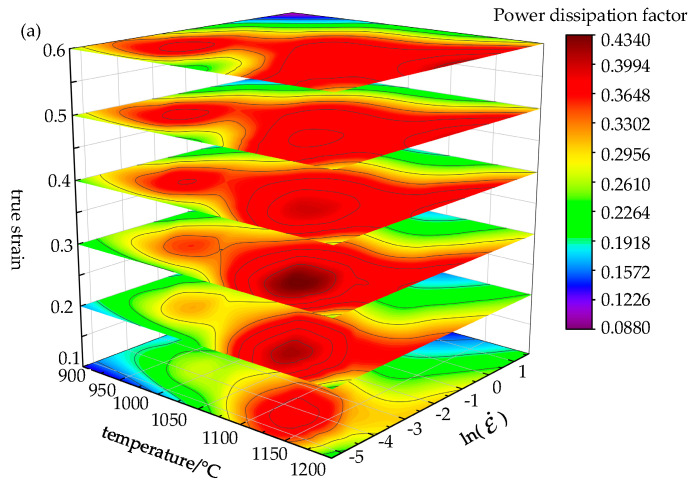
The 3D power dissipation map (**a**) and 3D instability map (**b**).

**Table 1 materials-14-02216-t001:** Chemical composition of Cr8 alloy steel (mass fraction %).

C	Si	Mn	Cr	Ni	Mo	V
0.48	0.54	0.6	7.64	0.48	1.56	0.13

**Table 2 materials-14-02216-t002:** Optimal parameters of the Hansel–Spittel constitutive model.

Parameter	*A*	*m_1_*	*m_2_*	*m_3_*	*m_4_*	*m_5_*	*m_7_*	*m_8_*	*m_9_*
Optimal Value	3.6 × 10^−10^	−0.00153	0.11594	−0.17799	−0.00576	9.28 × 10^−4^	−1.23057	3.11 × 10^−4^	−2.50901

## Data Availability

Data available on request due to restrictions eg privacy or ethical. The data presented in this study are available on request from the corresponding author. The data are not publicly available due to these data are part of ongoing research.

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
