# Peer review of "The Parameters Identification of High-Temperature Constitutive Model Based on Inverse Optimization Method and 3D Processing Map of Cr8 Alloy Steel"

_materials, 2021, doi:10.3390/ma14092216_

Round 1

Reviewer 1 Report

1. Please complete the information, what kind of lubricant was used in the upsetting process to minimize the friction in contact with the anvils?

2. Please provide information on the estimated time of cooling the sample after deformation and what was its temperature after cooling? The cooling time of the sample may have a significant effect on its structure, which may differ from the structure of the material under dynamic conditions. Then such structures will not correspond to real conditions. As a consequence, the analysis of the safe deformation areas will not correlate with the material structure and the conclusions from point 5 of the article.

3. line 173: is has three, better change to: can have three

4. Table. 1: Please comment the value of A = 3.6 x10 ^ 10 MPa. This quantity has no physical meaning as the material constant. Please comment in this light on the meaning of the remaining constants (m1… .m9) appearing in the equation (1).

5. In view of the above (point 4), the Hansel-Spittel model as a model of constitutive equations loses its physical sense. Please explain the usefulness of this model to describe the properties of the tested material. This problem is also reflected in the lack of accuracy of the approximation, especially visible in Fig. 6d and Fig. 6e. 

Reviewer's calculation example:

deformation 0.4, deformation speed 5 1/s, temperature 900 degrees Celsius; stress = 288 MPa (approx. 270 MPa in the Fig. 6a)

deformation 0.4, deformation speed 0,005 1/s, temperature 900 degrees Celsius; stress = 143 MPa (approx. 125 MPa in the Fig. 6e)

6. line 321: is: stain, is to be strain

7. There is no structural support for the analysis contained in item 5 of the article entitled Three-dimensional Hot Processing map of Cr8 Alloy Steel. Showing selected structures would prove the truthfulness of the analysis presented in the article. Please consider including selected structures in the article, although obtaining them may be difficult due to the remarks contained in p. 2 on the effect of the sample cooling time. 

Author Response

Dear reviewers, we have made some amendments as attached according to your review questions. Please check it out, thank you!

Reviewer 2 Report

In this article, The authors used the inverse optimization method to identify Hansel- Spittel constitutive model parameters of Cr8 alloy steel using compression test data for different temperatures, strain, and strain rates. They used the calibrated model to identify optimal thermo-mechanical processing parameters of Cr8 alloy steel and presented data in the form of 3D maps. The idea is generally exciting and helpful for the research community, but it needs clarifications and modifications before it can be accepted for publication in the Journal.

I did not understand that the authors talk about good deformation and wear resistance of this Cr8 alloy steel, which makes it a good candidate for the production of mill rolls, which should not deform during the forming process. A reader would assume that the authors would want to improve on these properties or at least identify the windows where the deformation of this material can be minimized. However, after reading the whole article, I came to realize that the authors did the opposite. They analyzed and identified the most efficient way of deforming this material and proposed windows of operation for this task. This is great for shaping this material in the desired form but does not comply with the material's niche, which has been mentioned several times in the article. It is essential to clarify this confusion and modify the appropriate text throughout the article

Major concerns:

  1. In Section 3, it has been clearly stated that only the compression tests were performed, but in the results section, it seems that the results for tensile tests have also been reported. Can you please elaborate on that?
  2. It would be great if the authors can fix the scales in figure 2 and figure 6 for a better comparison of flow curves at different strain rates. Showing the hardness curves on the secondary Y-axis would also help present the results better and comply with the explanation stated in the article body.
  3. The authors talk about the average grain size in different defamation cases on page 6. Can you please provide average grain size data for each case?
  4. To my understanding, the presence of shear bands in compression tests is not uniform throughout the sample. It largely depends on the position at which the microstructure is observed. Some authors in the past also tried to pinpoint the zones free of shear bands in the compression test samples. Can you please state precisely at which point these micrographs were recorded?
  5. On page 7, the authors state that sequential quadratic programming is an excellent way of doing inverse optimization. As compared to what? Can you please mention some other methods here as well?
  6. I did not exactly understand the difference between figure 7 and Figure 8. I believe they both are the exact figures but presented in different ways. If it is correct, instead of repeating the data twice, please try to find a convenient way of showing it comprehensively once.

Minor suggestions:

  1. Several references are missing in the introduction section of the article
  2. The material and experimental method can be explained in slightly more detail
  3. Caption Figure 8 is not complete, and a reference is missing
  4. Please add a list of abbreviations, acronyms, and symbols before the references section start for the readers to clearly understand the notions used in the whole document
  5. The language of the article needs significant improvement. There are many grammatical and punctuation errors. It is difficult to understand some of the sentences as well, which need rephrasing. Some sections of the article need to be rephrased for a better explanation. It would be great if the authors can use a native English speaker's help to improve the language used.
  6. The title of the article can be improved. Please try to make it comprehensive and inclusive so that more readers can relate to it.
  7. Please follow the Journal guidelines for authors strictly while revising the article.
  8. It would also be great if the authors can cite more recent and related publications from the MDPI-Materials Journal during revision.

Please refer to the attached PDF document with handwritten comments for further reference.

I wish you all the best for the future.

Author Response

(The authors gave the same response as above.)

Round 2

Reviewer 2 Report

Thank you for revising the article and providing point-to-point feedback. I agree with all the modifications and wish you the best of luck.